# An aging-independent replicative lifespan in a symmetrically dividing eukaryote

Eric C Spivey[1,2†], Stephen K Jones Jr[1,2†], James R Rybarski[1], Fatema A Saifuddin[1], Ilya J Finkelstein[1,2,3*]

[1]Department of Molecular Biosciences, The University of Texas at Austin, Austin, United States; [2]Center for Systems and Synthetic Biology, The University of Texas at Austin, Austin, United States; [3]Institute for Cellular and Molecular Biology, The University of Texas at Austin, Austin, United States

**Abstract** The replicative lifespan (RLS) of a cell—defined as the number of cell divisions before death—has informed our understanding of the mechanisms of cellular aging. However, little is known about aging and longevity in symmetrically dividing eukaryotic cells because most prior studies have used budding yeast for RLS studies. Here, we describe a multiplexed fission yeast lifespan micro-dissector (multFYLM) and an associated image processing pipeline for performing high-throughput and automated single-cell micro-dissection. Using the multFYLM, we observe continuous replication of hundreds of individual fission yeast cells for over seventy-five generations. Surprisingly, cells die without the classic hallmarks of cellular aging, such as progressive changes in size, doubling time, or sibling health. Genetic perturbations and drugs can extend the RLS via an aging-independent mechanism. Using a quantitative model to analyze these results, we conclude that fission yeast does not age and that cellular aging and replicative lifespan can be uncoupled in a eukaryotic cell.

*For correspondence:
ifinkelstein@cm.utexas.edu

†These authors contributed equally to this work

Competing interests: The authors declare that no competing interests exist.

## Introduction

Aging is the progressive decrease of an organism's fitness over time. The asymmetric segregation of pro-aging factors (e.g., damaged proteins and organelles) has been proposed to promote aging in mitotically active yeast and higher eukaryotes (*Bufalino et al., 2013*; *Erjavec et al., 2008*; *Henderson et al., 2014*; *Katajisto et al., 2015*). In budding yeast, asymmetric division into mother and daughter cells ensures that a mother cell produces a limited number of daughters over its replicative lifespan (RLS) (*Mortimer and Johnston, 1959*). Aging mother cells increase in size, divide progressively more slowly, and produce shorter-lived daughters (*Mortimer and Johnston, 1959*; *Bartholomew and Mittwer, 1953*; *Kennedy et al., 1994*). Mother cell decline is associated with asymmetric phenotypes such as preferential retention of protein aggregates, dysregulation of vacuole acidity, and genomic instability (*Henderson et al., 2014*; *Aguilaniu et al., 2003*; *Saka et al., 2013*). By sequestering pro-aging factors in the mothers, newly born daughters reset their RLS (*Henderson et al., 2014*; *Aguilaniu et al., 2003*; *Kwan et al., 2013*). These observations raise the possibility that alternate mechanisms may be active in symmetrically dividing eukaryotic cells.

The fission yeast *Schizosaccharomyces pombe* is an excellent model system for investigating RLS and aging phenotypes in symmetrically dividing eukaryotic cells. Fission yeast cells are cylindrical, grow by linear extension, and divide via medial fission. After cell division, the two sibling cells each inherit one pre-existing cell tip (old-pole). The new tip is formed at the site of septation (new-pole). Immediately after division, new growth is localized at the old-pole end of the cell. Activation of growth at the new-pole cell tip occurs ~30% through the cell cycle (generally halfway through $G_2$). This transition from monopolar to bipolar growth is known as new end take-off (NETO)

**eLife digest** As the cells in our bodies age, their ability to carry out their normal processes also degrades. Ultimately, this causes tissues to deteriorate. How rapidly a cell ages depends on the genes encoded in its DNA, and can also be affected by certain drug treatments.

Cells reproduce by dividing to form two new cells. One common approach used to study cellular aging is to follow how genetic modifications and drug treatments alter how many offspring a cell produces before it dies. So far, most of these studies have been performed using budding yeast cells, which reproduce by dividing asymmetrically to form two differently sized cells. Little is known about aging in cells that divide symmetrically.

Another type of yeast, called fission yeast, divides symmetrically. Furthermore, many of the processes used by fission yeast cells to repair and replicate DNA are the same as those used in human cells. To study aging in fission yeast, Spivey, Jones et al. developed a "microfluidic" device and software tools that can image and track a large number of cells across their entire lifespans (which last for up to a week).

Unexpectedly, it appears that fission yeast do not age. Although cells died, their deaths were random and their chance of dying did not increase as the cells got older. Treating the cells with a drug called rapamycin lengthened their average lifespan, as did stabilizing their DNA. Treatments that impaired the ability of the cells to repair their DNA reduced the yeast's lifespan. However, in all cases, a cell's chance of death stayed constant as the cells got older.

One conclusion that could be drawn from the work of Spivey, Jones et al. is that cellular aging may be intrinsic to cells that divide asymmetrically – so budding yeast ages whilst fission yeast does not. Future studies will investigate this in more detail; for example, by looking at how "pro-aging factors" are segregated between dividing cells.

(*Mitchison and Nurse, 1985*; *Sveiczer et al., 1996*; *Martin and Chang, 2005*). Prior studies of fission yeast have yielded conflicting results regarding cellular aging. Several papers reported aging phenotypes akin to those observed in budding yeast (e.g., mother cells become larger, divide more slowly, and have less healthy offspring as they age) (*Erjavec et al., 2008*; *Barker and Walmsley, 1999*). However, a recent report used colony lineage analysis to conclude that protein aggregates are not asymmetrically distributed, and that inheriting the old cell pole or the old spindle pole body during cell division does not lead to a decline in cell health (*Coelho et al., 2013*). However, this report tracked the first 7–8 cell divisions of microcolonies on agar plates and thus could not observe the RLS of single cells (*Coelho et al., 2013*). The controversy between these studies may partially stem from the difficulty in tracking visually identical cells for dozens of generations.

Replicative lifespan assays require the separation of cells after every division. This is traditionally done via manual micro-dissection of sibling cells on agar plates, a laborious process that is especially difficult and error-prone for symmetrically dividing fission yeast. Extrinsic effects related to using a solid agar surface may confound observations made under these conditions (*Mei and Brenner, 2015*). Finally, recent work using high-throughput microfluidic devices to study individual budding yeast and bacterial cells (*Lee et al., 2012*; *Crane et al., 2014*; *Wang et al., 2010*; *Liu et al., 2015*; *Jo et al., 2015*; *Nobs and Maerkl, 2014*; *Tian et al., 2013*; *Huberts et al., 2014*; *Minc and Chang, 2010*) has shown that large sample sizes are needed to truly capture cellular lifespan accurately – populations less than ~100 cells do not reliably estimate the RLS (*Huberts et al., 2014*).

Here, we report the first high-throughput characterization of both RLS and aging in fission yeast. To enable these studies, we describe a microfluidic device—the multiplexed fission yeast lifespan microdissector (multFYLM)—and a software analysis suite that capture and track individual *S. pombe* cells throughout their lifespan. Using this platform, we present the first quantitative replicative lifespan study in *S. pombe*, settling a long-standing controversy regarding whether this organism undergoes replicative aging (*Erjavec et al., 2008*; *Barker and Walmsley, 1999*; *Coelho et al., 2013*; *Minois et al., 2006*). The RLS of fission yeast is substantially longer than previously reported (*Erjavec et al., 2008*; *Barker and Walmsley, 1999*). Remarkably, cell death is stochastic and does not exhibit the classic hallmarks of cellular aging. Despite the lack of aging phenotypes, rapamycin

and Sir2p overexpression both extend RLS, whereas increased genome instability decreases RLS. These results demonstrate that RLS can be extended without any aging phenotypes, a fact that has implications for the proposed mechanism of lifespan extension by these interventions. Using these results, we also describe a quantitative framework for analyzing how stochastic and age-dependent effects contribute to the experimentally measured replicative lifespan. This framework will be broadly applicable to future replicative lifespan studies of model organisms. We conclude that fission yeast dies primarily via a stochastic, age-independent mechanism.

## Results

### A high-throughput microfluidic assay for measuring the replicative lifespan of fission yeast

Replicative lifespan assays require the separation of cells after every division. This is traditionally done via manual micro-dissection of sibling cells, a laborious process that is especially difficult for symmetrically dividing fission yeast. We recently developed a microfluidic platform for capturing and immobilizing individual fission yeast cells via their old cell tips (*Spivey et al., 2014*). However, our first-generation device could only observe a single strain per experiment, required an unconventional fabrication strategy, and suffered from frequent cell loss that ultimately shortened the observation time. To address these limitations, we developed a multiplexed fission yeast lifespan microdissector (multFYLM, *Figure 1*), along with a dedicated software package designed to streamline the analysis and quantification of raw microscopy data. Single cells are geometrically constrained within catch channels, preserving the orientation of the cell poles over multiple generations (*Figure 1A–B*). The cells divide by medial fission, thereby ensuring that the oldest cell pole is retained deep within the catch channel. If new-pole tips are loaded initially, then these outward-facing tips become the old-pole tips after the first division.

The multFYLM consists of six closely spaced, independent microfluidic subsystems, with a total capacity of 2352 cells (*Figure 1C*, *Figure 1—figure supplement 1*). To ensure equal flow rates throughout the device, each of the six subsystems is designed to have the same fluidic resistance. Each subsystem consists of eight parallel rows of 49 cell catch channels (*Figure 1D*). The catch channel dimensions were optimized for loading and retaining wild type fission yeast cells (*Figure 1E*, *Figure 1—figure supplement 2*; (*Spivey et al., 2014*). The eight rows of cell catch channels are arranged between a large central trench (40 µm W x 1 mm L x 20 µm H) and a smaller side trench (20 µm W x 980 µm L x 20 µm H). Cells are drawn into the catch channels by flow from the central trench to the side trenches via a small (2 µm W x 5 µm L x ~12 µm H) drain channel (*Figure 1F*). We did not monitor all 2352 catch channels during our experiments, but we routinely filled >80% of the 224 monitored catch channels in each of the six microfluidic sub-systems (*Figure 1—figure supplement 2A*). A constant flow of fresh media supplies nutrients, removes waste, and ensures that cells are stably retained for the duration of the experiment. Once loaded, up to 98% of the cells could be retained in their respective catch channels for over 100 hr (*Figure 1—figure supplement 2B–C*, *Video 1*). As a cell grows and divides, the old-pole cell is maintained by flow at the end of the catch channel, while the new-pole cells grow towards the central trench. Eventually, the new-pole cells are pushed into the central trench, where they are washed out by constant media flow (*Video 1*). This allows for continuous, whole-lifetime observation of the old-pole cell as well as its newest siblings (*Figure 1G–H*).

An inverted fluorescence microscope with a programmable motorized stage allowed the acquisition of single cell data with high-spatiotemporal resolution (*Figure 1—figure supplement 3A*). Following data collection, images are processed through the image analysis pipeline (see Supplemental Methods), which measures the length and fluorescence information for each cell at each time point (*Figure 1—figure supplement 3B*). These data are parsed to determine the cell's size, division times, growth rates, lifespan, and fluorescence information (if any). Cells within the multFYLM grew with kinetics and morphology similar to cells in liquid culture and showed growth and NETO rates (*Figure 2—figure supplement 1*, *Supplementary file 1A*) similar to those reported previously (*Coelho et al., 2013*; *Nobs and Maerkl, 2014*; *Forsburg and Rhind, 2006*). In sum, the multFYLM allows high content, continuous observation of individual fission yeast cells over their entire lifespans.

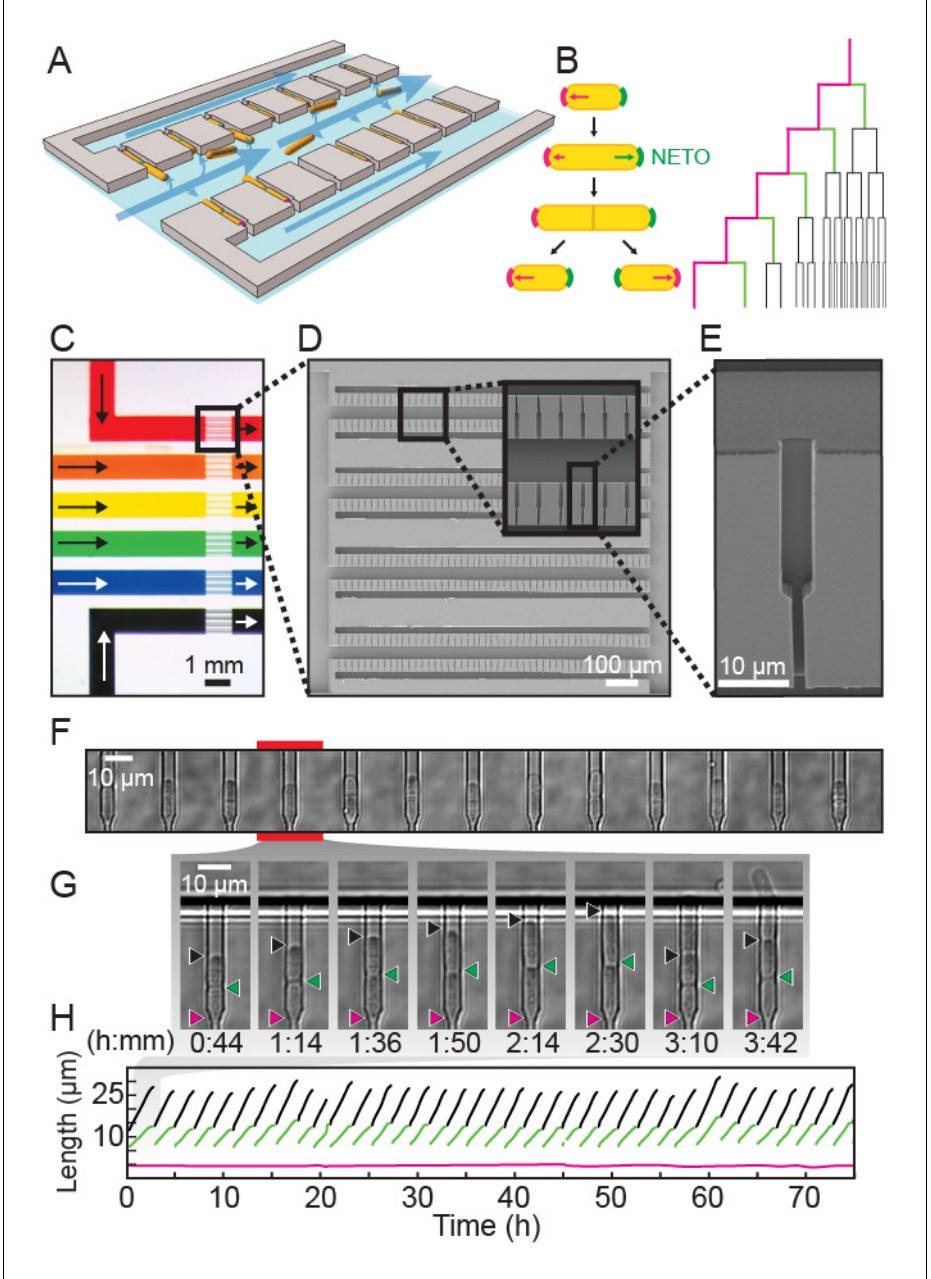

**Figure 1.** A multiplexed fission yeast lifespan microdissector (multFYLM). (**A**) Illustration of multFYLM (gray) loaded with fission yeast (orange). Blue arrows represent the media flow though the multFYLM. (**B**) Left: Fission yeast cells initially grow from the old-pole end (magenta). After new end takeoff (NETO), growth begins at the new-pole end (green). Right: multFYLM permits tracking of the old-pole cell, as well as its most recent siblings. (**C**) Optical image of a multFYLM showing six independent subsystems. Arrows indicate direction of media flow. Scanning electron micrographs of (**D**) a multFYLM subsystem and (**E**) a single catch channel. The channel is long enough to accommodate the old-pole cell, as well as the most recent new-pole sibling. (**F**) White-light microscope image of a row of catch channels loaded with cells. (**G**) Time-lapse images (**H**) and single-cell traces of a replicating cell. The old-pole (magenta) is held in place while the new-pole (green) is free to grow. The old-pole of the most recent sibling (black) extends until it is removed by flow into the central trench (after ~2 hr 30 m).

The following figure supplements are available for figure 1:

**Figure supplement 1.** Schematic of the multiplexed fission yeast lifespan microdissector (multFYLM).

**Figure supplement 2.** Loading and retention of cells in the multFYLM.

*Figure 1 continued on next page*

*Figure 1 continued*

**Figure supplement 3.** Experimental apparatus and image processing workflow.

## The fission yeast replicative lifespan is not affected by aging

We used the multFYLM to measure the fission yeast RLS (*Figure 2*). From these data, we plotted the replicative survival curve (*Figure 2A*) and determined the survival function, $S(g)$, which describes the probability of being alive after generation $g$. Using the survival data, we also computed the hazard function, $\lambda(g) = -\frac{dS(g)}{dg} * S(g)^{-1}$, which describes the instantaneous risk of death after cell division $g$ (*Figure 2B*). The hazard rate (also called the death rate) can be calculated for any generational age using this function. Surprisingly, the fission yeast survival curve did not fit the traditional aging-dependent Gompertz model, (*Gompertz, 1825*; *Greenwood, 1928*; *Wilson, 1993*), which describes survival and hazard in terms of a generation-dependent (*i.e.,* aging) and a generation-independent term (*Equation (2)* in Materials and methods). The RLS data were best described by a single exponential decay, corresponding to a generation-independent hazard rate. Strikingly, the hazard rate does not increase as the replicative age increases; instead, it remains steady at an average ~2% chance of death per cell per generation. For comparison, we also analyzed the survival data and hazard function for budding yeast (*S. cerevisiae*) grown in a microfluidic device (*Jo et al., 2015*). As expected, the budding yeast hazard function increases with each generation and fits the aging-dependent Gompertz model. Thus, the replicative age, $g$, strongly influences the probability of death in budding yeast, but not in fission yeast.

The RLS is defined by the number of generations at which 50% of the starting cells are dead. We measured an RLS of 39.2 generations (95% C.I. 38.6–39.8, n = 440, *Figure 2A* and *Supplementary file 1*) for the laboratory strain h- 972 grown in rich media (*Leupold, 1970*). To explore the effect of different genetic backgrounds on replicative lifespan, we determined survival curves, hazard functions, and replicative half-lives for three additional strains, including two wild fission yeast isolates with distinct morphologies and population doubling times (*Jeffares et al., 2015*). Most laboratory fission yeast strains have three chromosomes, but the recently identified strain CBS2777 has a fourth chromosome that originated via a series of complex genomic rearrangements (*Brown et al., 2014*, *2011*; *Rhind et al., 2011*). CBS2777 had the shortest RLS of 22.7 generations (95% CI 21.9–23.6 generations), consistent with its aberrant genome and altered genome maintenance (*Brown et al., 2014*). In contrast, strains NCYC132 and JB760 showed greater longevity with estimated RLS of >50 and >70 generations, respectively (*Figure 2—figure supplement 2*, *Supplementary file 1B*). These RLS correspond to a hazard rate that is ~5 fold lower than strain h- 972 (*Figure 2—figure supplement 2B*). The longevity of strain NCYC132 is consistent with an earlier manual microdissection study performed on solid media (*Coelho et al., 2013*). Each strain's survival curve was best described by an exponential decay function, resulting in a generation-independent hazard function. Thus, an aging-independent replicative lifespan is a feature of diverse fission yeasts isolates, and likely the entire species.

## Aging-associated phenotypes do not correlate with death in fission yeast

Prior RLS studies have identified three common phenotypes associated with cellular aging (*Henderson et al., 2008*; *Piper, 2006*). In budding yeast, aging mother cells increase in cell size, progressively slow their doubling times, and produce daughters with decreased fitness

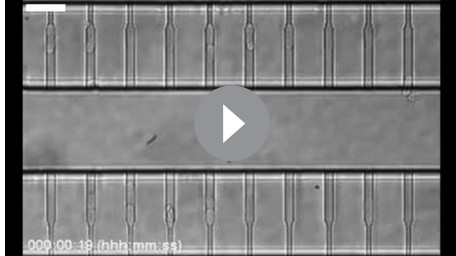

**Video 1.** Operation of the multFYLM. Time-lapse imaging of fission yeast in a single field of view of the multFYLM for approximately 140 hr. Scale bar is 20 μm.

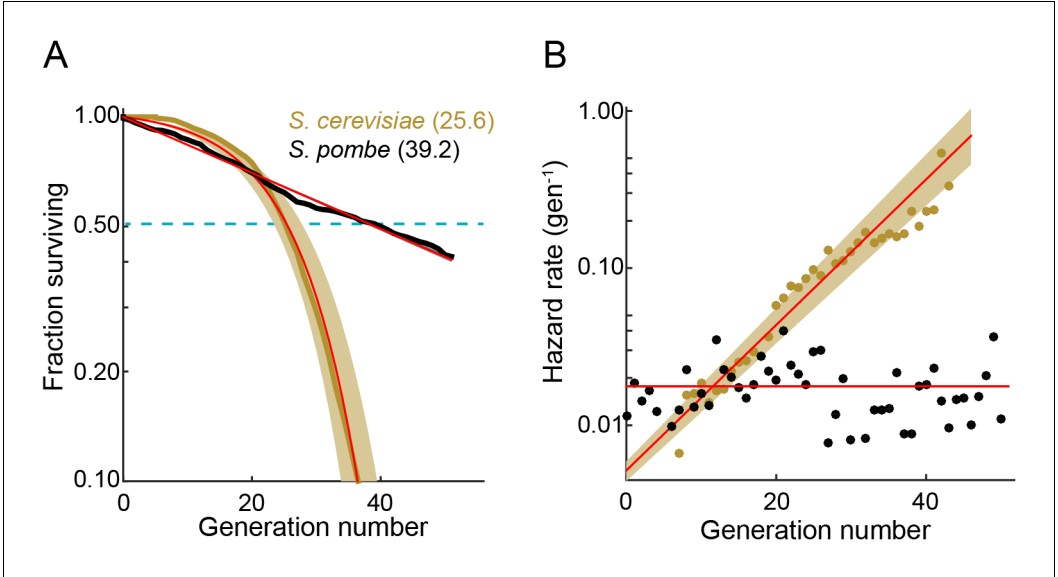

**Figure 2.** The fission yeast replicative lifespan (RLS). (**A**) Survival curves for wild-type *S. pombe* (black) and wild-type *S. cerevisiae* (brown, data from *Jo et al., 2015*); both were grown in microfluidic microdissection devices. Numbers indicate the average lifespan. Red lines are a fit to a Gompertz (*S. cerevisiae*) and exponential decay (*S. pombe*) survival models. Shading indicates 95% confidence interval (C.I.). Dashed blue line: 50% survival. (**B**) Hazard rate curves for the data shown in (**A**). The hazard rate increases dramatically with increased replicative age for *S. cerevisiae* but not for *S. pombe*.

The following figure supplements are available for figure 2:

**Figure supplement 1.** Health of cells in the multFYLM.
**Figure supplement 2.** Survival and hazard curves for wild type fission yeast isolates.

(*Mortimer and Johnston, 1959*; *Bartholomew and Mittwer, 1953*; *Kennedy et al., 1994*). Whether older fission yeast cells also undergo similar aging-associated phenotypes remains unresolved. To answer these outstanding questions, we examined time-dependent changes in morphology, doubling times, and sibling health in individual fission yeast cells as they approached death. Quantitative observation of cell length revealed two classes of dying cells: the majority (72%; n = 234) died prior to reaching the normal division length (<16.2 μm, *Figure 3A–B*). The remaining cells (28%; n = 92) showed an elongated phenotype and exceeded the average division time by more than three-fold. In *S. pombe,* cell length is strongly correlated with division time and cell length, indicating that cell cycle checkpoints were likely dysregulated in these dying cells (*Mitchison and Nurse, 1985*; *Sveiczer et al., 1996*; *Wood and Nurse, 2015*; *Hachet et al., 2012*). Despite these differences in length at death, most cells had normal doubling times throughout their lifespan (*Figure 3C–D*). Next, we investigated whether there were any morphological changes in the generations preceding death. The majority of cells retained wild type cell lengths and doubling rates until the penultimate cell division. In the two generations immediately preceding death, there was a mild, but statistically significant change in the distributions of cell lengths (*Figure 3E*) and doubling times (*Figure 3F*). However, we did not observe any predictable trends for individual cells, arguing against a consistent pattern of age-related decline. In sum, *S. pombe* dies without any aging-associated morphological changes.

We next determined the fate of the last sibling produced by dying cells. This is possible because the last siblings of dying cells also remain captured in the catch channels (*Figure 4A*). These are siblings that are produced by the final division of the old-pole cells. We categorized these siblings into three classes: (i) those that died without dividing, (ii) those that divided once and died, and (iii) those that divided two or more times (*Figure 4A*, *Videos 2–4*). The fate of the new-pole siblings was

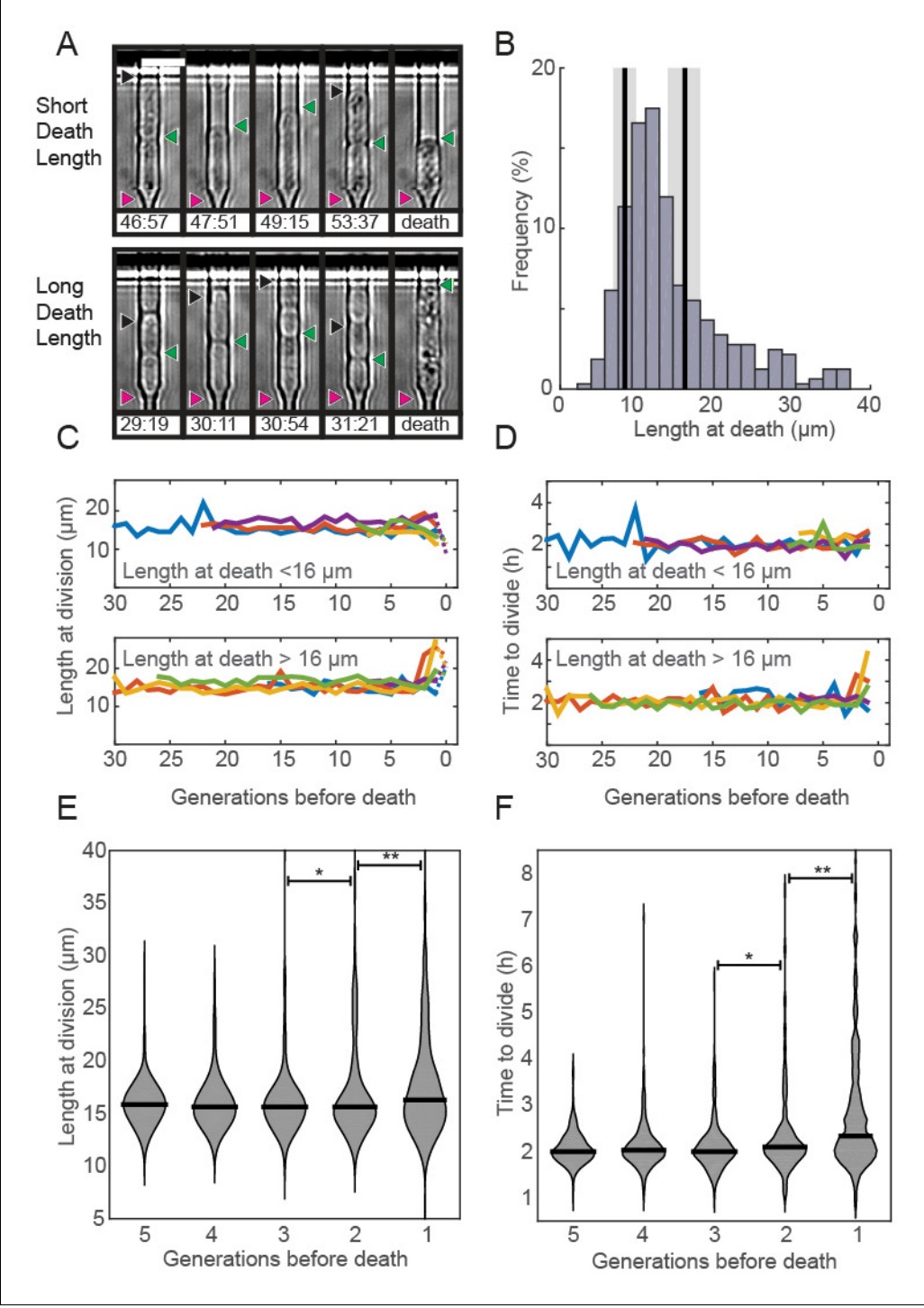

**Figure 3.** Fission yeast does not show signs of aging. (**A**) Images of cells showing a short (top) and long (bottom) phenotype at death. Triangles indicate the old-pole, new-pole, and the new-pole of the previous division as in *Figure 1G*. Scale bar: 10 μm. (**B**) Histogram of cell length at death. The birth length was 8.3 ± 1.5 μm (mean ± st. dev., n = 326) and the length at division was 16 ± 2.2 μm (n = 326). (**C**) Cells dying with either short (top) or long (bottom) phenotype have normal length and (**D**) doubling times prior to death, as indicated by five representative cells. (**E**) Distribution of length and (**F**) doubling time at division in the five generations preceding death. Cells were post-synchronized to the time of death. The black bar shows the median value for each generation (n > 290 cells for all conditions). Sequential generations were compared using the Kolmogorov-Smirnov test (* for p<0.05, ** for p<0.01).

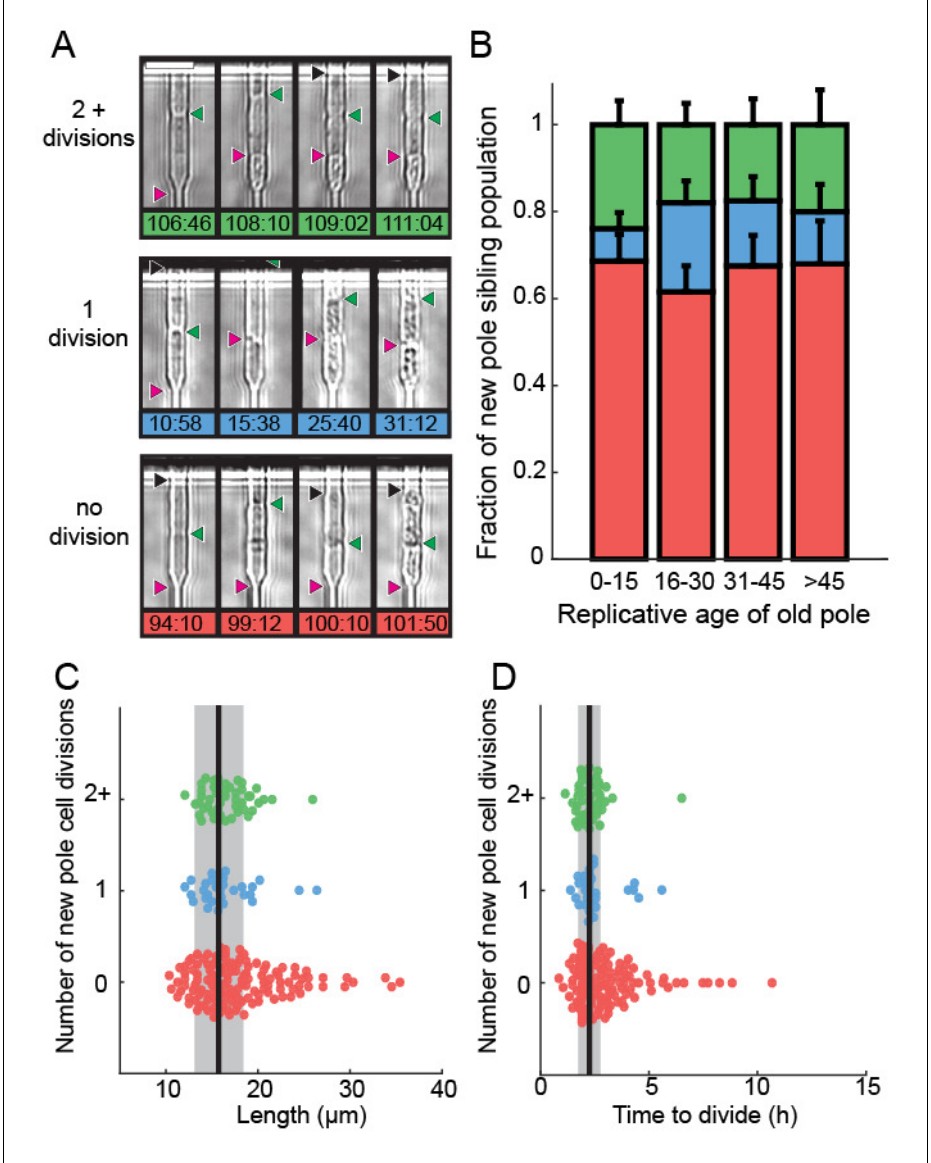

**Figure 4.** Analysis of siblings born during the last division of a dying cell. (A) Last new-pole sibling continued to divide (top), died after one division (middle) or died without dividing (bottom). (B) The distribution of last-sibling phenotypes as a function of the old-pole replicative age (n = 245). Error bars are st. dev. measured by bootstrap analysis. (C) The length at division and (D) the doubling time of the new-pole siblings. Vertical black lines and gray bars show the mean and standard deviation of the total cell population.

independent of the replicative age of the old-pole cells at death (*Figure 4B*). Most new-pole siblings of a dying cell (66%, n = 135) never divided and typically did not grow, suggesting that the underlying cause of death was distributed symmetrically between the two sibling cells. Similarly, new-pole siblings that divided once and died (14%, n = 29) also typically did not grow. old-pole cells that died while hyper-elongated were more likely to have these unhealthy offspring (*Figure 4C–D*). As elongated cells generally indicate the activation of a DNA damage checkpoint (*Furnari et al., 1997*), our findings suggest that genome instability in these cells may be the underlying cause of death in both the old cell and its most recent sibling. In new-pole siblings that divided two or more times (20%, n = 40), doubling times were indistinguishable from rapidly growing, healthy cells (2.1 ± 0.7 hr, n = 208). These healthy new-pole cells were typically born from old-pole cells that did not elongate

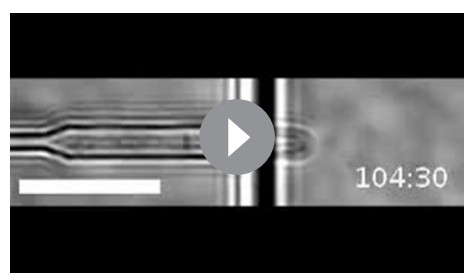

**Video 2.** New-pole sibling phenotypes. Time-lapse images of a cell where the last division produces a healthy new-pole cell that divides more than once. Scale bar is 20 µm.

during their terminal division (*Figure 4C–D*). These observations suggest that the cause of death was contained within the old-pole cell, thereby reducing the likelihood of death in the sibling. Taken together, our observations suggest that in 76% of cases, cell death impacts both sibling cells. However, in 24% of cells, death is localized to just one of the two siblings. Importantly, there was no correlation between the replicative age of the old-pole cell and survival probability of the last sibling cell. These observations are in stark contrast to *S. cerevisiae,* where aging factors are generally sequestered to younger mother cells (*Henderson et al., 2014*; *Zhou et al., 2014*). However, older *S. cerevisiae* mothers produce larger and slower-dividing daughter cells. Thus, we do not observe any aging-dependent outcomes for the fate of the new-pole sibling, further confirming that *S. pombe* does not age.

## Genetic manipulation and rapamycin treatment extend replicative lifespan

The histone deacetylase Sir2p modulates lifespan and aging in a wide variety of organisms from yeasts to mice (*Wierman and Smith, 2014*; *Donmez and Guarente, 2010*; *Ganley and Kobayashi, 2014*). For example, *sir2* deletion (*sir2Δ*) in budding yeast reduces the replicative lifespan by ~50% (*Jo et al., 2015*; *Kaeberlein et al., 1999*), whereas Sir2p overexpression can increase the RLS by up to 30% (*Kaeberlein et al., 1999*). The *S. pombe* genome encodes three Sir2p homologs, one of which shares a high degree of sequence similarity and biochemical functions with the budding yeast Sir2p (*Shankaranarayana et al., 2003*; *Freeman-Cook et al., 2005*). Although *S. pombe* does not show aging phenotypes, we next investigated whether Sir2p still modulates the RLS. Deletion of Sir2p (*sir2Δ)* reduced the RLS by 15% (33.4 ± 0.8 generations; n = 329) relative to the wild type parental strain (39.2 ± 0.6 generations, n = 440; *Figure 5A*). These results are consistent with prior observations that wild type and *sir2Δ* cells had similar growth rates when cultured without stressors (*Erjavec et al., 2008*). In contrast, constitutive 2-fold *sir2* overexpression (*sir2OE, Figure 5—figure supplement 1*) increased the RLS over 50%, with a mean replicative lifespan of >60 generations (n = 301; *Figure 5A*). Cells overexpressing Sir2p have a 2-fold lower hazard rate (total risk of death) than wild-type cells (*Figure 5B* and *Figure 5—figure supplement 2*). In sum, overexpression of Sir2p increases the RLS of *S. pombe*, but does not affect aging.

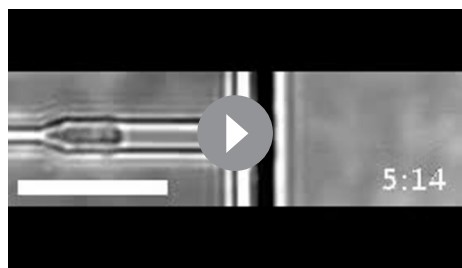

**Video 3.** New-pole sibling phenotypes. Time-lapse images of a cell where the last division produces an unhealthy new-pole cell that divides once before dying. Scale bar is 20 µm.

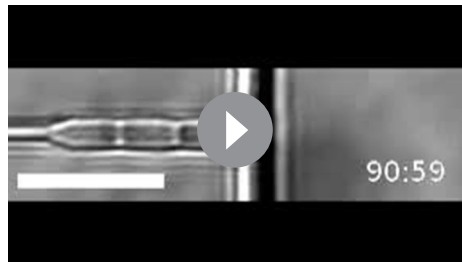

**Video 4.** New-pole sibling phenotypes. Time-lapse images of a cell where the last new-pole cell does not divide. Scale bar is 20 µm.

The target of rapamycin (TOR) pathway is the central regulator of cell growth and aging in eukaryotes (*Johnson et al., 2013*; *Loewith and Hall, 2011*). Rapamycin inhibits TOR proteins in eukaryotes (*Kunz et al., 1993*; *Brown et al., 1994*), reducing cell growth by altering translation initiation and repressing ribosome biogenesis. Rapamycin increases the RLS of budding yeast (*Medvedik et al., 2007*), and also increases the longevity of worms, flies and mice (*Robida-Stubbs et al., 2012*; *Bjedov et al., 2010*; *Harrison et al., 2009*). However, the effect of rapamycin on fission yeast RLS has not been reported. The addition of 100 nM rapamycin to the flow medium increased the RLS by 41% (55.3 generations, 95% CI 53.1–57.6, n = 184; *Figure 5A*). As with Sir2p overexpression, treating cells with rapamycin increased their longevity by reducing the aging-independent hazard rate (*Figure 5B*). Importantly, both rapamycin and Sir2p overexpression did not affect the lengths or doubling times of cells as they approached death. The cell length at division remained constant for the five generations preceding cell death, whereas the cell doubling time increased only in the last generation before death (*Figure 5C–F*). These results are consistent with an aging-independent mechanism for RLS extension in *S. pombe*.

## Ribosomal DNA (rDNA) instability contributes to sudden cell death

Ribosomal DNA (rDNA) is a repetitive, recombination-prone region in eukaryotic genomes. Defects in rDNA maintenance arise from the illegitimate repair of the rDNA locus and have been proposed to be key drivers of cellular aging (*Ganley and Kobayashi, 2014*; *Kobayashi, 2008*). Fission yeast encodes ~117 rDNA repeats on the third chromosome, and incomplete rDNA segregation can activate the spindle checkpoint (*Toda et al., 1984*; *Maleszka and Clark-Walker, 1993*). If unresolved, this can result in chromosome fragmentation and genomic instability (*Win et al., 2005*). Indeed, we observed that 28% of dying cells were abnormally long, suggesting activation of a DNA-damage checkpoint (*Figure 3B*). Aberrant rDNA structures are processed by the RecQ helicase Rqh1p and mutations in the human Rqh1p homologs are implicated in cancer and premature aging-associated disorders (*Croteau et al., 2014*; *Ellis et al., 1995*; *Yu et al., 1996*; *Kitao et al., 1999*; *Takahashi et al., 2011*). Therefore, we sought to determine whether rDNA instability and loss of Rqh1p can contribute to stochastic death.

We visualized segregation of rDNA in cells using the nucleolar protein Gar2 fused to mCherry. Gar2 localizes to rDNA during transcription and has been used to monitor rDNA dynamics in live cells (*Win et al., 2005*). The strain expressing Gar2-mCherry at the native locus divided at approximately the same length and rate as wild-type cells (*Figure 6—figure supplement 1*). Dividing cells mostly had equal Gar2 fluorescence in both sibling cell nuclei (*Figure 6A–B*). However, 7% of cells (n = 88/1182) showed rDNA segregation defects that appear as multiple rDNA loci, asymmetric fluorescence distributions, or rDNA bridges (*Figure 6A*). The rDNA mis-segregation was similar to mis-segregation of a fluorescent marker on chromosome I (7% of cells; n = 20/278) and chromosome II (5% of cells; n = 19/370) (*Figure 6—figure supplement 2*). Multi-punctate rDNA loci were nearly always lethal, whereas asymmetric rDNA segregation and rDNA bridges were lethal in 40% of cells (n = 31/78; *Figure 6B*). Importantly, 40% of cells (n = 56/141) showed rDNA defects immediately preceding death. In dying cells, rDNA instability was elevated relative to mis-segregation of chromosome I (16% of cells; n = 15/95) and chromosome II (13% of cells; n = 18/137) (*Figure 6—figure supplement 2*). Loss of Rqh1p caused higher frequencies of spontaneous rDNA defects, a shorter RLS (5.3 generations, 95% CI 5.0–5.7), and a >7 fold higher hazard rate (*Figure 6—figure supplement 3*). rDNA defects were even more prevalent in dying *rqh1Δ* cells (*Figure 6—figure supplement 3*). We next tested the influence of Sir2p over-expression or addition of rapamycin on the short RLS of *rqh1Δ* cells. The addition of rapamycin showed a mild RLS extension, whereas the *rqh1Δ sir2OE* strain was extremely short lived (*Figure 6—figure supplement 3*). These data suggest that the elevated rDNA instability cannot be rescued by Sir2p and is only partially rescued by rapamycin. In summary, the longevity of fission yeast is promoted by rDNA stability.

## Discussion

Here, we report the first study of fission yeast RLS in a high-throughput microfluidic device (*Figure 1*). The multFYLM provides a temperature-controlled growth environment for hundreds of individual cells for up to a week (>75 generations), facilitating tens of thousands of micro-dissections. We also developed an image-processing pipeline for quantitative phenotypic analysis of individual

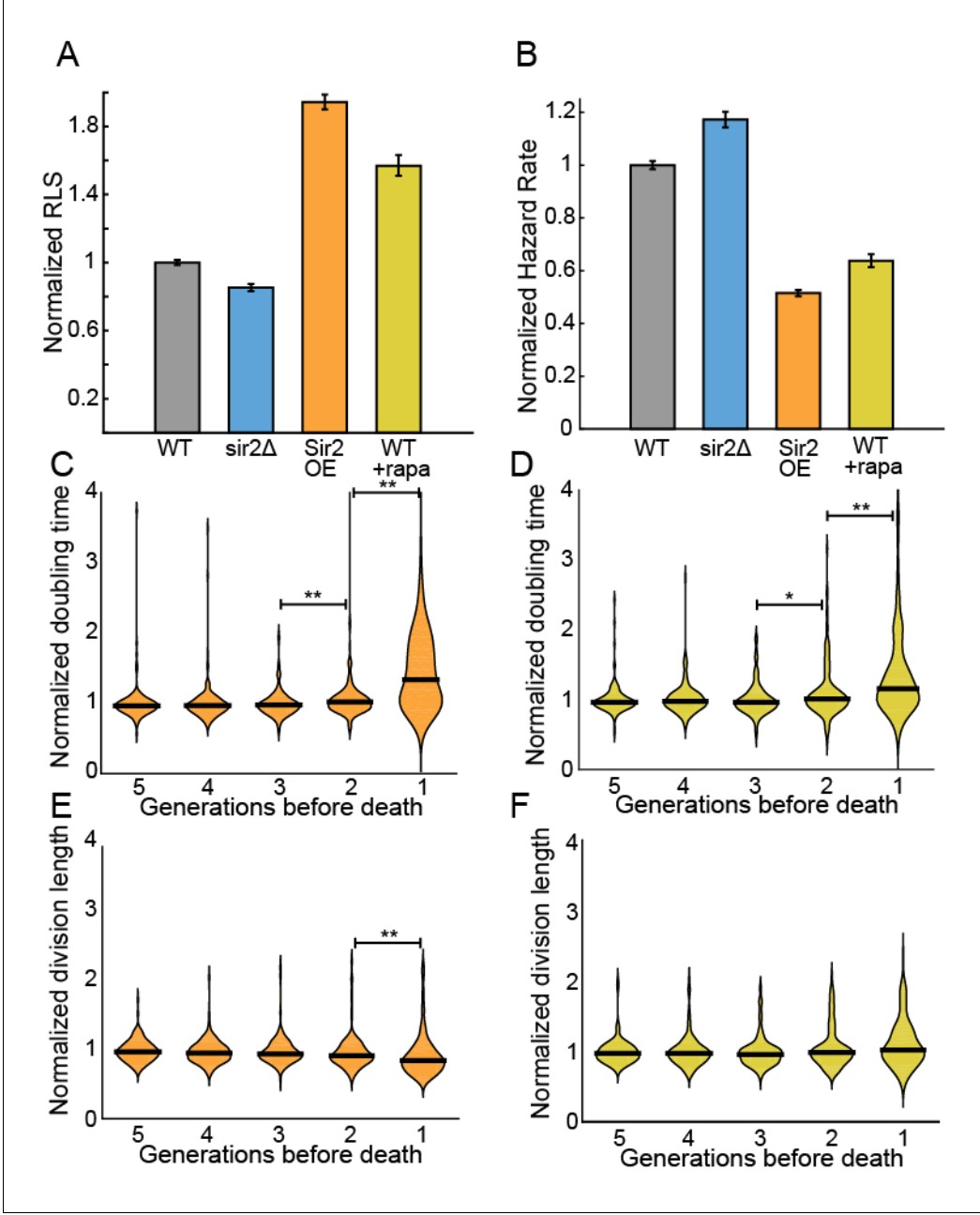

**Figure 5.** Sir2p and rapamycin extend replicative lifespan. (A) Replicative lifespans and (B) hazard rates of strains normalized to the mean RLS and hazard rate of wild-type (h- 972) strain. Error bars: 95% C.I. on an exponential decay fit to the experimental survival curve. (C) Distribution of normalized doubling time in the five generations preceding death for Sir2p overexpression cells and (D) wild-type cells treated with 100 nM rapamycin. (E) Distribution of normalized length at division in the five generations preceding death for Sir2p overexpression cells and (F) wild-type cells treated with 100 nM rapamycin. Black bars show the median value. Sequential generations were compared using the Kolmogorov-Smirnov test (* for p<0.05, ** for p<0.01; n > 144 cells for all conditions).

The following figure supplements are available for figure 5:

**Figure supplement 1.** Sir2p expression levels.

**Figure supplement 2.** The effect of Sir2p and rapamycin on the RLS of fission yeast.

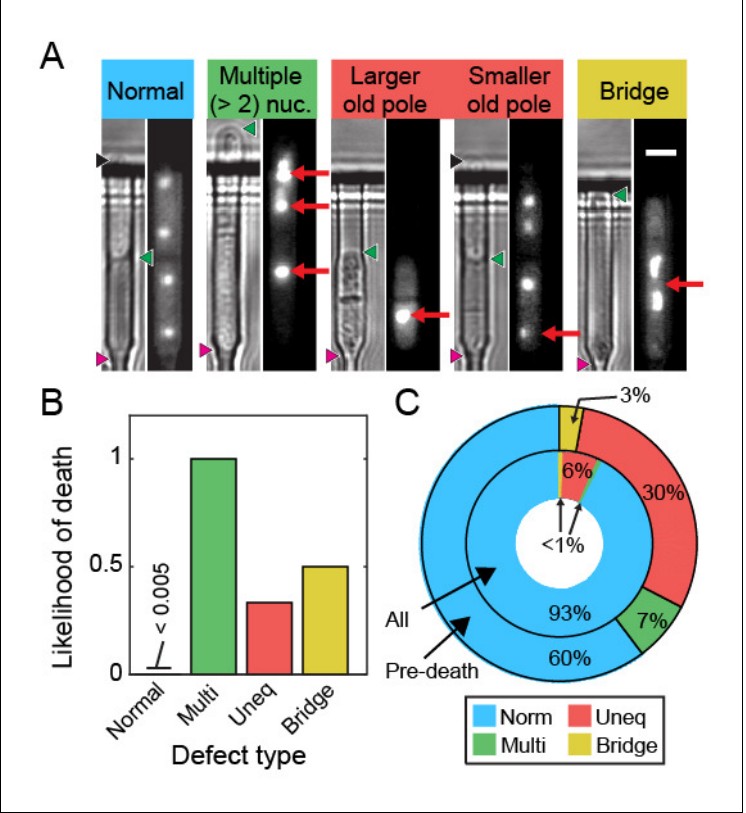

**Figure 6.** Ribosomal DNA (rDNA) instability is highly correlated with cell death. (**A**) Images of cells exhibiting rDNA instability, as reported by gar2-mCherry, which binds to rDNA. (**B**) Likelihood of cell death following one of the defects observed in (**A**). (**C**) Dying cells exhibited elevated rDNA defects (outer ring) relative to healthy dividing cells (inner ring).

The following figure supplements are available for figure 6:

**Figure supplement 1.** Strains expressing gar2-mCherry maintain wild-type replication rates.

**Figure supplement 2.** Live imaging of chromosome mis-segregation rates.

**Figure supplement 3.** Characterization of the RLS of an *rqh1Δ* strain.

cell lineages. The blueprints for multFYLM, as well as the image analysis pipeline are both available via GitHub (see Materials and methods). We note that microfluidics-based lifespan measurements are free from potential extrinsic effects (e.g., secretion of small molecules onto the solid agar surface) which have been proposed to confound observations of RLS assays using manual microdissection (*Mei and Brenner, 2015*). Using the multFYLM, we set out to determine whether fission yeast undergoes replicative aging.

Taken together, our data provide three sets of experimental evidence that fission yeast does not age. First, the RLS survival curves and corresponding hazard rates are qualitatively different between budding and fission yeasts (*Figure 2*). The budding yeast RLS is best described by a Gompertz model that includes both an age-dependent as well as an age-independent survival probability. In contrast, the fission yeast RLS is best described by a single exponential decay. This corresponds to a hazard function $\lambda(g)$ that is constant with each generation, as would be expected for an organism that does not age. Second, we observed that cell volumes and division rates did not change until the penultimate cell division (*Figure 3*). Third, after the death of a cell, the health of the surviving sibling was also independent of age (*Figure 4*). These conclusions are also broadly consistent with a

recent report that observed robust, aging-independent growth of the symmetrically dividing *E. coli* in a microfluidic device (*Wang et al., 2010*).

Our results highlight a critical consideration when interpreting replicative lifespan curves. The experimentally determined RLS does not necessarily report on whether a population of cells is aging. This is because the RLS is affected by a combination of both age-independent and age-dependent biological processes—cell may die randomly without aging. A mathematical model is required to further parse the relative contributions of aging-independent and aging-dependent mechanisms. The Gompertz function is an excellent model for understanding the replicative lifespan survival curve (*Bansal et al., 2015*). In this model, the replicative lifespan is dependent on just two parameters: (i) an age-independent hazard rate, and (ii) an age-dependent hazard rate, which determine how rapidly mortality increases as a function of the cell's age (*Kirkwood, 2015*). Together, these two parameters describe a population's likelihood of death at any age. *Figure 7* illustrates that the experimentally observed RLS is dependent on both of these terms. An increased RLS is not sufficient to conclude that cells age more slowly. Indeed, our data indicate that a decreased RLS can also be achieved by increasing the age-independent hazard rate. To further illustrate this point, we include a Gompertz model analysis of the RLS of *S. cerevisiae* grown in a microfluidic device (*Jo et al., 2015*) and our own data from *S. pombe*. For all of our data, changes in RLS can be completely accounted for by changes in the age-independent hazard rate. In contrast, budding yeast have age-dependent hazard rates an order of magnitude higher than their age-independent rates (*Figure 7*). We anticipate that the development of the multFYLM, as well as a quantitative analysis framework, will continue to inform comparative RLS studies in model organisms.

What are the molecular mechanisms of stochastic cell death in a symmetrically dividing eukaryote? Strikingly, the RLS of *S. pombe* can be extended by up to 41% via rapamycin treatment and Sir2p over-expression without influencing aging-related phenotypes (*Figure 5*). This suggests that rapamycin and Sir2p both reduce the onset of sudden death. To our knowledge, this is the first evidence these interventions can extend the RLS in an organism that does not age. This suggests that

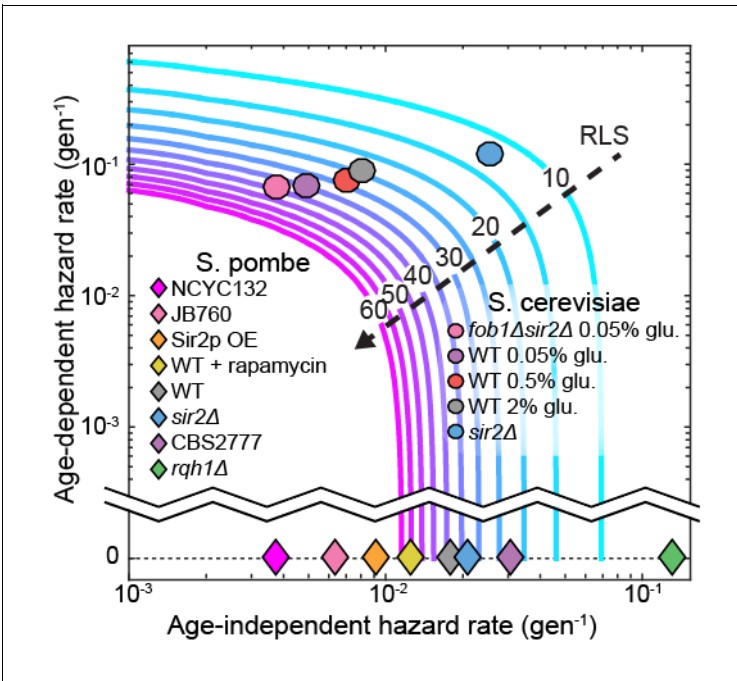

**Figure 7.** The replicative lifespan is an incomplete reporter of cellular aging. RLS contours were generated from experimentally determined ranges of Gompertz coefficients using *Equation (7)*. Fission yeast strains examined in this study and budding yeast from an analogous study (*Jo et al., 2015*) were plotted on the chart based on the coefficient values from either a Gompertz or exponential decay fit. In all cases, the 95% CI of the coefficients was smaller than the marker size.

both Sir2p and rapamycin may partially affect the RLS of aging organisms like *S. cerevisiae* via an aging-independent mechanism. Additional studies will be required to parse out the aging-dependent and aging-independent mechanisms of RLS extension.

For a subset of cells, genomic instability due to aberrant segregation of the rDNA locus may be one cause of death (*Saka et al., 2013*; *Ganley and Kobayashi, 2014*). Consistent with this hypothesis, rDNA segregation defects were highly elevated relative to general chromosome mis-segregation in wild type cells immediately prior to death (*Figure 6*). Ablating Rqh1p, a helicase that promotes rDNA stability and replication fork progression (*Stewart et al., 1997*; *Murray et al., 1997*; *Doe et al., 2000*), further increased rDNA defects while drastically reducing the RLS, independent of age. Recent studies also indicate that protein aggregation may be a second major cause of sudden death in rapidly dividing fission yeast cells (*Coelho et al., 2013*, *2014*). These two mechanisms of death need not be mutually exclusive. Future studies will further define the molecular mechanisms of aging-independent death in fission yeast.

## Materials and methods

### multFYLM master structure photolithography

Master structures were fabricated using SU-8 3005 photoresist for the first layer and SU-8 2010 photoresist for the second layer, following standard photolithography techniques described in the product literature (Microchem, Westborough, MA). Photoresist was spun onto 100 mm-diameter, test-grade, P-doped silicon wafers (ID# 452, University Wafers). Photoresist thickness was 20–30 µm for the second layer (conduits, central and side trenches), and 5–6 µm for the first layer (catch and drain channels). Custom chrome on quartz photomasks (Compugraphics) were manufactured from designs created using the freeware integrated circuit layout editor Glade (www.peardrop.co.uk) or with OpenSCAD (www.openscad.org). A Suss MA-6 Mask Aligner (Suss MicroTec Lithography GmbH) was used for mask alignment and photoresist exposure.

### multFYLM construction

multFYLM construction is described in more detail at Bio-protocol (*Jones Jr et al., 2018*). Approximately 25 g of polydimethylsiloxane (PDMS, Sylgard 184, Dow Corning) was mixed at a weight ratio of 10:1 polymer:hardener, placed on a rotator for >30 min, then centrifuged to remove bubbles. A tape barrier was applied to the edge of a silicon wafer bearing the multFYLM master structures, and 13 g of PDMS was poured onto the wafer, covering the surface. The wafer with PDMS was degassed for ~10 min (at 60–70 cmHg vacuum in a vacuum chamber) to remove additional bubbles, then placed in a 70°C oven for 15–17 min. The wafer was intentionally removed while the PDMS was still tacky to improve adhesion of the microfluidic connectors (nanoports). To create a source and drain interface for the device, nanoports (N-333–01, IDEX Health and Science, Oak Harbor, WA) were applied to the surface of the PDMS over the ends of the master structure of the multFYLM, then an additional 14 g of PDMS was poured over the surface, with care taken to avoid getting liquid PDMS inside the nanoports. The wafer with PDMS and nanoports was then degassed >10 min to remove additional bubbles, and then returned to the 70°C oven for 3 hr to cure completely. The cured PDMS was then removed whole from the wafer after cooling to room temperature. A multFYLM with nanoports was trimmed from the cured PDMS disk to approximately 15 × 25 mm. A 1 mm diameter biopsy punch (Acu-punch, Accuderm) was used to make holes connecting the bottom of each nanoport to the conduits on the bottom surface of the PDMS. The multFYLM was then placed in isopropanol and sonicated for 30 min, removed, and placed to dry on a 70°C hotplate for 2 hr. Borosilicate glass coverslips (48 × 65 mm #1, Gold Seal) were concurrently prepared by cleaning for >1 hr in a 2% detergent solution (Hellmannex II, Hellma Analytics), then rinsing thoroughly in deionized water and isopropanol before drying for >2 hr on a 70°C hotplate. Cleaned multFYLM and coverslips were stored in a covered container until needed. To bond, the multFYLM and a coverslip were placed in a plasma cleaner (PDC-32G, Harrick Scientific) and cleaned for 20 s on the 'high' setting with oxygen plasma from ambient air (~21% oxygen). The multFYLM and glass coverslip were then gently pressed together to form a permanent bond. To make the microfluidic interface, PFA tubing (1512L, IDEX), with a 1/16" outer diameter was used to connect a 100 mL luer lock syringe (60271, Veterinary Concepts, www.veterinaryconcepts.com) to the multFYLM, and to

construct a drain line leading from the multFYLM to a waste container. The tubing was connected to the multFYLM nanoports using 10–32 coned nuts (F-332–01, IDEX) with ferrules (F-142N, IDEX). A two-way valve (P-512, IDEX) was placed on the drain line to allow for better flow control. For longer experiments, two syringes could be loaded in tandem, connected with a Y-connector (P-512, IDEX). The valve and Y-connector were connected to the tubing using ¼−28 connectors (P-235, IDEX) with flangeless ferrules (P-200, IDEX). The source line tubing was connected to the syringe with a Luer adapter (P-658, IDEX). The syringe(s) was/were then placed in a syringe pump (Legato 210, KD Scientific) for operation. A 2 µm inline filter (P-272, IDEX) was placed upstream of the multFYLM to reduce the chance of debris clogging the multFYLM during operation.

## Loading the multFYLM with cells

As soon as possible after plasma bonding (usually within 15 min), the multFYLM was placed on the microscope stage and secured, then 15 µL of cells suspended in YES media with 2% BSA were gently injected into the source nanoport. The drain and source interface tubing (with drain valve closed) were then connected to the drain and source nanoports. The syringe pump was then started at 40 µL min$^{-1}$, and the drain valve was opened to allow flow to start. The syringe pump was adjusted between 20–60 µL min$^{-1}$ until flow was established, and catch channels were observed to be filling with cells, then a programmed flow cycle was established: 1–5 min at 50 µL min$^{-1}$ followed by 10–14 min at 5 µL min$^{-1}$ (average flow rate, 0.5–1.2 mL h$^{-1}$).

## Time-lapse imaging

Images were acquired using an inverted microscope (Nikon Eclipse Ti) running NIS Elements and equipped with a 60X, 0.95 NA objective (CFI Plan Apo λ, Nikon) and a programmable, motorized stage (Proscan III, Prior). The microscope was equipped with Nikon's 'Perfect Focus System' (PFS), which uses a feedback loop to allow consistent focus control over the multiple-day experiments. Images were acquired approximately every two minutes using a scientific-grade CMOS camera (Zyla 5.5, Andor). To improve contrast during image analysis, images were acquired both in the plane of focus and 3–4 µm below the plane of focus. Temperature control was maintained using an objective heater (Bioptechs) and a custom-built stage heater (Omega) calibrated to maintain the multFYLM at 30–31°C.

Fluorescence images were acquired with a white light LED excitation source (Sola II, Lumencorp) and a red filter set (49004, Chroma). Fluorescent images were acquired concurrently with white light images, but only every four minutes.

## Media and strains

All strains were propagated in YES liquid media (Sunrise Scientific) or YES agar (2%) plates, except where otherwise noted. A complete list of strains is reported in *Supplemental file 1*. Deletion of *sir2* in the wild-type h- 972 strain was completed by first generating a PCR product containing KANMX4 flanked by *SIR2* 5' and 3' untranslated regions. Competent IF30 cells were then transformed with the PCR product and selected on YES +G418 agar plates. Deletion was confirmed by PCR. A *SIR2* over-expression plasmid was generated via Gateway cloning (Life Technologies). First, *SIR2* was integrated into the pDONR221 plasmid to yield pIF133. *SIR2* was then swapped from pIF133 to pDUAL-GFH1 (Riken), yielding plasmid pIF200. pIF200 expresses *SIR2* under the *NMT1* promoter, and also labels the protein C-terminally with a eGFP-FLAG-His6 tags. Plasmid pIF200 was then transformed into strain IF140 for integration at the *leu1* locus to generate clones IF230 and IF231 (*Matsuyama et al., 2006*). Integration was checked by PCR and expression was determined by RT-qPCR. Wild-type h- 972 and Gar2-mCherry fusion protein expressing strains (Megan King) were transformed with a PCR product containing *HPHMX6* (hygromycin resistance) flanked by *rqh1* 5' and 3' untranslated regions. Transformants were selected on YES+hygromycin agar plates, then confirmed by PCR. Chromsome I and II reporter strains were kindly provided by Christian Haering (strains CH2774 and CH3245) (*Petrova et al., 2013*).

## Western blotting

Strains IF235 (*sir2Δ*), IF140 and IF230 (*sir2* over-expression; *sir2OE*) were grown in synthetic complete media (Sunrise Scientific) to OD 1.0, and whole cell extracts were prepared by trichloroacetic

acid precipitation with bead lysis (**Keogh et al., 2006**). These cell extracts were combined with loading buffer, boiled for 5 min at 95°C, and resolved via a 4–20% TGX SDS-PAGE gel (Bio-Rad). Proteins were transferred to a PVDF membrane, and then blocked with PBS Odyssey blocking buffer (Li-Cor, Inc.) for one hour with shaking at 22°C. The membrane was incubated overnight at 4°C while shaking in phosphate buffered saline +0.05% Tween-20 (PBST) containing 1:1000 dilution of anti-$\beta$ actin antibody (mouse, ab8224, Abcam) and 1:500 dilution of an *S. pombe*-specific anti-Sir2p antibody (rabbit, generously provided by Dr. Allshire) (**Buscaino et al., 2013**). The membrane was washed once in PBST, and then agitated for 4 hr at 22°C with 1:10,000 dilution of goat anti-mouse IgG and 1:10,000 dilution of goat anti-rabbit IgG (IRDye 680RD and IRDye 800CW, respectively, Li-Cor, Inc.). The membrane was washed three times with PBST and imaged on an Odyssey CLx dual-channel imaging system (Li-Cor, Inc.).

## FYLM Critic: An open-source image processing and quantification package

To analyze our single-cell data, we developed FYLM Critic—a Python package for rapid and high-content quantification of the time-lapse microscopy data (github.com/finkelsteinlab/fylm). The FYLM Critic pipeline consists of two discrete stages: (i) pre-processing the raw microscope images to correct for stage drift, and (ii) quantification of cell phenotypes (e.g., length, doubling time, fluorescence intensity and spatial distribution).

## Stage 1: Pre-processing the microscope images

Several image pre-processing steps were taken to permit efficient quantification. First, the angles of the four solid sections of PDMS on either side of the catch channels were measured relative to the vertical axis of the image, and a corrective rotation in the opposite direction was applied to all images in that field of view. This resulted in the long axis of each cell being completely horizontal in all subsequent analyses. The multFYLM was too large to be imaged in its entirety within a single field of view, so the microscope stage was programmed to move to eight different subsections of the device in a continuous cycle. Due to the imperfect motion of the microscope stage, subsequent images of the same field of view randomly translated by a few micrometers relative to the previous image at the same nominal location. Images were aligned to the first image of the sequence using a cross-correlation algorithm (**Guizar-Sicairos et al., 2008**). The exact location of each catch channel was manually specified, but only if a cell had already entered the channel at the beginning of the experiment.

## Stage 2: Annotating individual cells within the multFYLM

In order to analyze cell lengths and phenotypes, kymographs were produced for each catch channel and then annotated semi-manually. To produce the kymographs, a line of pixels was taken along the center of each catch channel, starting from the center of the 'notch' and extending into the central trench. This kymograph captured information along the long axis of the cells. The time-dependent one-dimensional (1D) kymographs for each catch channel were stacked together vertically, producing a two-dimensional (2D) kymograph of each cell's growth and division. The out-of-focus image was used for this process as the septa and cell walls were much more distinct, as described previously (**Nobs and Maerkl, 2014**). The position of the old-pole and the leftmost septum were then manually annotated using a simple point-and-click interface to identify each cell division. Once all single-cell kymographs were annotated, this annotation was used to calculate a cell length for each time point. The final status of the cell (whether it died, survived to the end of the experiment, or was ejected) was determined by the user. Further analysis of the cell length, growth rates, and fluorescence intensities was performed in MATLAB (Mathworks). The local minima of the growth curves were used to determine division times and the replicative age of each cell.

Ejections and replacement of individual cells within the catch channels may confound our whole-lifespan tracking. Therefore, we optimizing cell loading, retention and imaging conditions to minimize the loss of individual cells throughout the course of each experiment. First, ejection of individual cells was minimized over the course of ~100 hr (**Figure 1—figure supplement 2B**). Second, each cell was imaged once every two minutes, minimizing the amount of time during which a cell could conceivably be ejected and replaced by another from the central trench. If a cell were indeed

replaced within our two-minute frame rate, we would expect an abrupt change in the captured cell length (indicated a re-loading event). All kymographs were carefully monitored for such events, and in the rare cases where this occurred, were discarded from the final analysis. In summary, our analysis ensures that cell ejection or re-capture is mitigated by both efficient cell capture and by analysis of the single-cell datasets.

## Survival curves

Kaplan-Meier (*Kaplan and Meier, 1958*) survival function estimates were calculated in MATLAB. Due to the low number of lost cells (*Figure 1—figure supplement 2B*) and the highest total number of total cells (n $\geq$ 100 for all experiments), the survival curves did not include cells that were lost during the course of the experiment (i.e., there was no right censoring). Our analysis is consistent with similar studies in *S. cerevisiae* (*Crane et al., 2014*). The survival function estimates were fitted with either a simple exponential decay function

$$S(g) = e^{-\alpha * g} \tag{1}$$

or the Gompertz survival function (*Gompertz, 1825*; *Greenwood, 1928*)

$$S(g) = e^{\alpha/\beta\left(1 - e^{\beta g}\right)} \tag{2}$$

Where S is the fraction of the initial population surviving at generation $g$. The coefficients α and β are >0 and <1, and are further defined for the hazard function $\lambda(g)$ below. The plotted survival data were weighted at $1/S(g)$ to increase the influence of older cells on the fit.

The hazard function $\lambda(g)$ (the probability of cell death during a given generation) can be derived from the survival function:

$$\lambda(g) = \frac{-\dot{S}(g)}{S(g)} \tag{3}$$

where $\dot{S}$ is the first derivative of $S$. For a survival curve fit to an exponential decay function, the corresponding hazard function is a constant:

$$\lambda(g) = \alpha \tag{4}$$

For a survival curve fit to the Gompertz function, the hazard function simplifies to:

$$\lambda(g) = \alpha \cdot e^{\beta g} \tag{5}$$

Where the coefficient $\alpha$ amplitude-scales $\lambda(g)$, and so has an age-independent influence on the hazard function, while the coefficient $\beta$ time-scales $\lambda(g)$, and so provides a compact method to describe the age-dependent increase in the probability of cell death (*Figure 7*). Note that as $\beta \to 0$, *Equation (5)* approaches *Equation (4)*.

Mean replicative lifespan (RLS) can be calculated by setting $S(g) = \frac{1}{2}$, and solving for $g$ (*Simms, 1946*). For exponential decay, RLS is simply the half-life of the decay function:

$$RLS = ln(2)/\alpha \tag{6}$$

For the Gompertz function, RLS is:

$$RLS = \frac{\ln\left(1 + \left(\frac{\beta \ln(2)}{\alpha}\right)\right)}{\beta} \tag{7}$$

## Statistical data analysis

All fitting and statistical data analysis were performed in MATLAB. Briefly, datasets were first tested for normality using the Anderson-Darling method, and parametric tests were used when possible. The fitting of all data was performed in MATLAB using either the nonlinear least squares method or least squares method. Determination of whether a survival curve was fit to an exponential decay or Gompertz function was made primarily by selecting the fit with the highest adjusted $r^2$ value.

## Acknowledgements

We are indebted to our colleagues Jürg Bähler, Christian Haering, Megan King, Edward Marcotte, Akihisa Matsuyama, Ronit Weisman, and Blerta Xhemalce for valuable strains and reagents. We are grateful to Xenia Brianna Gonzalez for her assistance with data quantification, and to other members of the Finkelstein laboratory for carefully reading the manuscript. We thank our colleagues Edward Marcotte, Andreas Matouschek and Aashiq Kachroo for critical feedback. This work was supported by the American Federation of Aging Research (AFAR-020), the National Institute of Aging (F32 AG053051 to SKJ), CPRIT (R1214 to IJF), and the Welch Foundation (F-l808 to IJF). IJF is a CPRIT Scholar in Cancer Research. The content is solely the responsibility of the authors and does not necessarily represent the official views of the National Institutes of Health.

## Additional information

### Funding

| Funder | Grant reference number | Author |
| --- | --- | --- |
| American Federation for Aging Research | AFAR-020 | Eric C Spivey<br>Stephen K Jones Jr<br>James R Rybarski<br>Fatema A Saifuddin<br>Ilya J Finkelstein |
| National Institute on Aging | F32 AG053051 | Stephen K Jones Jr |
| Cancer Prevention and Research Institute of Texas | R1214 | James R Rybarski<br>Fatema A Saifuddin<br>Ilya J Finkelstein |
| Welch Foundation | F-l808 | Eric C Spivey<br>Stephen K Jones Jr<br>James R Rybarski<br>Fatema A Saifuddin<br>Ilya J Finkelstein |

The funders had no role in study design, data collection and interpretation, or the decision to submit the work for publication.

### Author contributions

ECS, SKJ, Conceptualization, Resources, Data curation, Software, Formal analysis, Supervision, Funding acquisition, Validation, Investigation, Visualization, Methodology, Writing—original draft, Project administration, Writing—review and editing; JRR, Resources, Software, Visualization, Methodology, Writing—original draft, Writing—review and editing; FAS, Data curation, Software, Validation, Visualization; IJF, Conceptualization, Formal analysis, Supervision, Funding acquisition, Investigation, Methodology, Writing—original draft, Project administration, Writing—review and editing

### Author ORCIDs

Eric C Spivey, http://orcid.org/0000-0002-4080-8616
Ilya J Finkelstein, http://orcid.org/0000-0002-9371-2431

## Additional files

### Supplementary files

• Supplementary file 1. This supplemental file contains tables of data used in making figures and also a table of strains used. It references four works not referenced in the main text: *Wood and Nurse, 2013*; *Xhemalce and Kouzarides, 2010*; *Weisman et al., 2005*; *Nakazawa et al., 2008*

### Major datasets

The following previously published datasets were used:

| Author(s) | Year | Dataset title | Dataset URL | Database, license, and accessibility information |
|---|---|---|---|---|
| Wood V, Gwilliam R, Rajandream MA, Lyne M, Lyne R, Stewart A, Sgouros J, Peat N, Hayles J, Baker S, Basham D, Bowman S, Brooks K, Brown D, Brown S, Chillingworth T, Churcher C, Collins M, Connor R, Cronin A, Davis P, Feltwell T, Fraser A, Gentles S, Goble A, Hamlin N, Harris D, Hidalgo J, Hodgson G, Holroyd S, Hornsby T, Howarth S, Huckle EJ, Hunt S, Jagels K, James K, Jones L, Jones M, Leather S, McDonald S, McLean J, Mooney P, Moule S, Mungall K, Murphy L, Niblett D, Odell C, Oliver K, O'Neil S, Pearson D, Quail MA, Rabbinowitsch E, Rutherford K, Rutter S, Saunders D, Seeger K, Sharp S, Skelton J, Simmonds M, Squares R, Squares S, Stevens K, Taylor K, Taylor RG, Tivey A, Walsh S, Warren T, Whitehead S, Woodward J, Volckaert G, Aert R, Robben J, Grymonprez B, Weltjens I, Vanstreels E, Rieger M, Schäfer M, Müller-Auer S, Gabel C, Fuchs M, Düsterhöft A, Fritzc C, Holzer E, Moestl D, Hilbert H, Borzym K, Langer I, Beck A, Lehrach H, Reinhardt R, Pohl TM, Eger P, Zimmermann W, Wedler H, Wambutt R, Purnelle B, Goffeau A, Cadieu E, Dréano S, Gloux S, Lelaure V, Mottier S, Galibert F, Aves SJ, Xiang Z, Hunt C, Moore K, Hurst SM, Lucas M, Rochet M, Gaillardin C, Tallada VA, Garzon A, Thode G, Daga RR, Cruzado L, Jimenez J, Sán- | 2002 | The genome sequence of Schizosaccharomyces pombe | https://www.ncbi.nlm.nih.gov/nuccore/CU329670.1 | Publicly available at the NCBI Nucleotide (accession no: CU329670.1) |

| | | | | |
|---|---|---|---|---|
| chez M, del Rey F, Benito J, Domínguez A, Revuelta JL, Moreno S, Armstrong J, Forsburg SL, Cerutti L, Lowe T, McCombie WR, Paulsen I, Potashkin J, Shpakovski GV, Ussery D, Barrell BG, Nurse P | | | | |
| Wood V, Gwilliam R, Rajandream MA, Lyne M, Lyne R, Stewart A, Sgouros J, Peat N, Hayles J, Baker S, Basham D, Bowman S, Brooks K, Brown D, Brown S, Chillingworth T, Churcher C, Collins M, Connor R, Cronin A, Davis P, Feltwell T, Fraser A, Gentles S, Goble A, Hamlin N, Harris D, Hidalgo J, Hodgson G, Holroyd S, Hornsby T, Howarth S, Huckle EJ, Hunt S, Jagels K, James K, Jones L, Jones M, Leather S, McDonald S, McLean J, Mooney P, Moule S, Mungall K, Murphy L, Niblett D, Odell C, Oliver K, O'Neil S, Pearson D, Quail MA, Rabbinowitsch E, Rutherford K, Rutter S, Saunders D, Seeger K, Sharp S, Skelton J, Simmonds M, Squares R, Squares S, Stevens K, Taylor K, Taylor RG, Tivey A, Walsh S, Warren T, Whitehead S, Woodward J, Volckaert G, Aert R, Robben J, Grymonprez B, Weltjens I, Vanstreels E, Rieger M, Schäfer M, Müller-Auer S, Gabel C, Fuchs M, Düsterhöft A, Fritzc C, Holzer E, Moestl D, Hilbert H, Borzym K, Langer I, Beck A, Lehrach H, Reinhardt R, Pohl TM, Eger P, Zimmermann W, Wedler H, Wambutt R, Purnelle B, Goffeau A, Cadieu E, Dréano S, Gloux S, Lelaure V, Mottier | 2002 | The genome sequence of Schizosaccharomyces pombe | https://www.ncbi.nlm.nih.gov/nuccore/CU329671.1 | Publicly available at the NCBI Nucleotide (accession no: CU329671.1) |

S, Galibert F, Aves SJ, Xiang Z, Hunt C, Moore K, Hurst SM, Lucas M, Rochet M, Gaillardin C, Tallada VA, Garzon A, Thode G, Daga RR, Cruzado L, Jimenez J, Sánchez M, del Rey F, Benito J, Domínguez A, Revuelta JL, Moreno S, Armstrong J, Forsburg SL, Cerutti L, Lowe T, McCombie WR, Paulsen I, Potashkin J, Shpakovski GV, Ussery D, Barrell BG, Nurse P

| | | | | |
|---|---|---|---|---|
| Wood V, Gwilliam R, Rajandream MA, Lyne M, Lyne R, Stewart A, Sgouros J, Peat N, Hayles J, Baker S, Basham D, Bowman S, Brooks K, Brown D, Brown S, Chillingworth T, Churcher C, Collins M, Connor R, Cronin A, Davis P, Feltwell T, Fraser A, Gentles S, Goble A, Hamlin N, Harris D, Hidalgo J, Hodgson G, Holroyd S, Hornsby T, Howarth S, Huckle EJ, Hunt S, Jagels K, James K, Jones L, Jones M, Leather S, McDonald S, McLean J, Mooney P, Moule S, Mungall K, Murphy L, Niblett D, Odell C, Oliver K, O'Neil S, Pearson D, Quail MA, Rabbinowitsch E, Rutherford K, Rutter S, Saunders D, Seeger K, Sharp S, Skelton J, Simmonds M, Squares R, Squares S, Stevens K, Taylor K, Taylor RG, Tivey A, Walsh S, Warren T, Whitehead S, Woodward J, Volckaert G, Aert R, Robben J, Grymonprez B, Weltjens I, Vanstreels E, Rieger M, Schäfer M, Müller-Auer S, Gabel C, Fuchs M, Düsterhöft A, Fritzc C, Holzer E, Moestl D, Hilbert H, Borzym K, Langer I, | 2002 | The genome sequence of Schizosaccharomyces pombe | https://www.ncbi.nlm.nih.gov/nuccore/CU329672.1 | Publicly available at the NCBI Nucleotide (accession no: CU329672.1) |

Beck A, Lehrach H, Reinhardt R, Pohl TM, Eger P, Zimmermann W, Wedler H, Wambutt R, Purnelle B, Goffeau A, Cadieu E, Dréano S, Gloux S, Lelaure V, Mottier S, Galibert F, Aves SJ, Xiang Z, Hunt C, Moore K, Hurst SM, Lucas M, Rochet M, Gaillardin C, Tallada VA, Garzon A, Thode G, Daga RR, Cruzado L, Jimenez J, Sánchez M, del Rey F, Benito J, Domínguez A, Revuelta JL, Moreno S, Armstrong J, Forsburg SL, Cerutti L, Lowe T, McCombie WR, Paulsen I, Potashkin J, Shpakovski GV, Ussery D, Barrell BG, Nurse P

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
