## [Decision Letter]

[Editors’ note: this article was originally rejected after discussions between the reviewers, but the authors were invited to resubmit after an appeal against the decision.]

Thank you for submitting your work entitled "An aging-independent replicative lifespan in a symmetrically dividing eukaryote" for consideration by *eLife*. Your article has been favorably evaluated by a Senior Editor and three reviewers, one of whom is a member of our Board of Reviewing Editors.

Our decision has been reached after consultation between the reviewers. Based on these discussions and the individual reviews below, we regret to inform you that your work will not be considered further for publication in *eLife*.

Each reviewer thought the study was technically well done and the results showing that *S. pombe* does not age were convincing. However, the major issue comes down to novelty. Two of the reviewers correctly pointed out that the microfluidic assay has already been published from your lab, and another group previously showed that *S. pombe* does not age, but using a microcolony-based assay. Your new high-throughput system is much better, but the conclusion that *S. pombe* does not age was already published, making the current findings less novel. After extensive debate, we decided the story does not reach the bar for novelty to be acceptable for publication *eLife*. I'm sorry for the disappointing news. The individual reviewer comments are included below.

*Reviewer #1:*

This manuscript by Spivey et al. describes a novel high-throughput microfluidic system for tracking the replicative lifespan (RLS) of the symmetrically dividing yeast, *Schizosaccharomyces pombe*. There have been conflicting reports of whether *S. pombe* actually ages or not, probably because of the difficulty in reliably following individual cells that have divided through fission. The multiplexed fission yeast lifespan microdissector (FYLM) solves this problem by forming a trap that traps and retains one cell pole and then allows new cell poles from the fission to flow away. The authors also developed a robust image analysis software package with the clever name "FYLM Critic". An earlier, single-channel version of this microfluidic system was previously described by the authors as a proof of principal, but this one appears to be much improved. FYLM is used in this study to demonstrate that *S. pombe* cells do not display age-associated changes in growth or morphology, and their rate of death does change with replicative age. Rather than fitting a Gompertz model for survival that takes into account age-dependent and age-independent contributions, the *S. pombe* data instead fits a model of single exponential decay. These cells apparently die stochastically, without any increased risk per generation.

Sir2 overexpression and rapamycin treatment, two conditions that extend *S. cerevisiae* RLS and lifespan of other model eukaryotes, actually extended *S. pombe* RLS as well. However, the effects remained independent of aging, and instead worked through aging-independent mechanisms. Deletion of the SIR2 gene had no effect on RLS, which was somewhat surprising given the positive effect of overexpression. SIR2 mRNA levels were increased 6-fold as measured by RT-PCR. However, this does not necessarily mean that the Sir2 protein level increased that much. Sir2 is epitope tagged in this strain, so the authors should look at Sir2 protein by western blotting.

mCherry-tagged Gar2 was used to monitor rDNA locus dynamics/segregation in the live cells. There is appears to be a correlation between improper rDNA segregation and RLS. Furthermore, a very short-lived *rqh1∆* helicase mutant displayed even more missegregation. The authors therefore conclude that Rqh1p promotes longevity by suppressing rDNA instability. However, the extremely short RLS of the *rqh1∆* mutant could be caused by additional, more general chromosome segregation defects. Some kind of control for general chromosome segregation seems important, perhaps centromeres.

Sir2-dependent rDNA array stabilization is considered one of the key mediators of RLS regulation in *S. cerevisiae*. It would be interesting to test whether Sir2 overexpression or rapamycin prevents the rDNA dynamics defects in replicatively "aging" *S. pombe* cells. In other words, is the rDNA contributing to the aging-independent RLS extension by Sir2 overexpression or rapamycin?

*Reviewer #2:*

This manuscript investigates replicative lifespan (RLS) in *S. pombe* using a microfluidics device developed by the Finkelstein lab. This allows them to follow individual cells up to 75 divisions before more than 50% of a population of cells has died. The conclusions from the study are that *S. pombe* does not "age", i.e. the cells do not show the expected features commonly associated with aging, like a slower division time and larger cell volume in the last few divisions before death. Furthermore, the likelihood of death does not increase with increasing numbers of cell divisions. Also, the siblings of dead cells do not show an increased chance of death. The system is further characterized to recapitulate the fact that Sir2 overexpression and rapamycin treatment increase RLS, and that increasing rDNA recombination causes decreased lifespan.

The conclusions of this study are congruent with an earlier study from the lab of I. Tolic-Norrelykke (Coelho, Current Biology 2013), which also concluded that *S. pombe* does not age. The present work follows individual cells over a much longer period of divisions, whereas the Coelho study analysed microcolonies with up to eight cell divisions. Nonetheless, the conclusions are basically very similar.

This study extends the observations of Coelho by analysis of Sir2 overexpression and rapamycin, two interventions that were previously known to extend lifespan.

From all I can tell, the data collection and analysis is sound. The novelty lies in the extent of data collection and the statistics, but the findings support earlier conclusions. Hence, it will be a judgment call as to whether this constitutes enough novelty for publication in *eLife*.

*Reviewer #3:*

The authors used a microfluidic device to trap individual fission yeast cells and monitored their growth and death patterns. The main conclusion is that fission yeast does not age, in terms of growth and death rates, with respect to replicative age. While the manuscript presents interesting, solid results, the microfluidic device ("FYLM") was published by the same group previously (Spivey et al., 2014) and the conclusion that *S. pombe* does not age was also published by another group (Coelho et al., 2013). The major contribution of the present work is the confirmation of the latter by extending the replicative age in the experiments by 5 to 10 fold. These recent advancements are reminiscent of the development in bacterial senescence several years ago (Wang et al., 2010 and references therein).

Comments:

1) In Abstract, "Nearly all RLS studies have used budding yeast[…]" -> "Nearly all RLS studies of single-cell eukaryotes have used budding yeast[…]"

2) Also in Abstract, "Here, we describe a multiplexed fission yeast lifespan micro-dissector (FYLM); a microfluidic platform for performing high-throughput and automated single-cell micro- dissection." This sentence reads as if this is the first time the authors are reporting their microfluidic device, but it was already published in Spivey et al., 2014.

3) The puns "FYLM", "multiFYLM", "FYLM Critic" are cute, but rather disingenuous. The Figure 1 in the manuscript and the figure in the Abstract in Spivey et al., 2014 remarkably resembles Figure 1 in Wang et al., 2010 for bacteria. Why invent a new name when the basic design of the device is essentially identical to what has been already published and widely adopted?

4) Figure 2. I suggest to plot from 0.1 to 1.0 for y-axis. Since the main result in this figure is the exponential decay of 'fraction surviving', the current range (0.01 to 1.0) puts unnecessary importance on what is less important.

5) Figure 2 and in the main text. I suggest the authors call "the hazard function" a more easily understandable and straightforward "death rate".

6) Figure 2—figure supplement 2; Figure 5—figure supplement 2. The exponential decay and constant death rates are not as clear here. For 2B, the data is presented in a misleading manner. The range of y axis should be from 0 to 0.07. The red fit lines should not mask the data in both Figure 2 and Figure 2.

---

## [Author Response]

[Editors’ note: the author responses to the first round of peer review follow.]

*[…] Reviewer #1:*

*[…] Sir2 overexpression and rapamycin treatment, two conditions that extend S. cerevisiae RLS and lifespan of other model eukaryotes, actually extended S. pombe RLS as well. However, the effects remained independent of aging, and instead worked through aging-independent mechanisms. Deletion of the this SIR2 gene had no effect on RLS, which was somewhat surprising given the positive effect of overexpression. SIR2 mRNA levels were increased 6-fold as measured by RT-PCR. However, this does not necessarily mean that the Sir2 protein level increased that much. Sir2 is epitope tagged in this strain, so the authors should look at Sir2 protein by western blotting.*

We clarified the results in the subsection “Genetic manipulation and rapamycin treatment extend replicative lifespan”, to state that the *sir2Δ* RLS is 15% shorter than that of wild type (*wt*) cells (also see Supplementary file 2). This effect is significantly smaller than reported in *S. cerevisiae* but mirrors an earlier observation that the *S. pombe sir2Δ* strain retains nearly the same fitness as wild type cells (Erjavec et al., 2008, DOI: 10.1073/pnas.0804550105). The 40-50% reduction of the *S. cerevisiae* RLS in a *sir2Δ* background may stem from the key role that Sir2p plays in maintaining genetic stability at the rDNA locus (Saka et al., 2013, DOI: 10.1016/j.cub.2013.07.048). In contrast, *S. pombe* appears to have a redundant, Sir2p-independent mechanism for maintaining rDNA stability (Shankaranarayana et al., 2003, DOI: 10.1016/S0960-9822(03)00489-5).

A revised supplemental figure (Figure 5—figure supplement 1) now includes additional characterization of Sir2p protein levels (Western blot) and localization. We used an antibody raised against the *S. pombe* Sir2p (graciously provided by Dr. Allshire (Buscaino et al., 2013, DOI: 10.1038/emboj.2013.72)) to blot both Sir2p-eGPF in the *LEU1* locus (clones IF230, IF231) as well as the native Sir2p expressed from its endogenous promoter. The antibody detected both native Sir2p and Sir2p-eGFP (seen as a slower-migrating species in panel B). Importantly, the antibody did not show any signal in the *sir2Δ* background. Panel (C) of the revised figure also demonstrates that Sir2p-eGFP localizes to the nucleus. Taken together, these results confirm ~2-3 fold over-expression of Sir2p in this construct. This data is also mentioned in the aforementioned subsection of the revised manuscript.

*mCherry-tagged Gar2 was used to monitor rDNA locus dynamics/segregation in the live cells. There is appears to be a correlation between improper rDNA segregation and RLS. Furthermore, a very short-lived rqh1∆ helicase mutant displayed even more missegregation. The authors therefore conclude that Rqh1p promotes longevity by suppressing rDNA instability. However, the extremely short RLS of the rqh1∆ mutant could be caused by additional, more general chromosome segregation defects. Some kind of control for general chromosome segregation seems important, perhaps centromeres.*

We investigated the rate of chromosome mis-segregation by monitoring the dynamics of fluorescently marked chromosome loci (Figure 6—figure supplement 2). Chromosome loci were marked with a TetO cassette that was integrated into Chromosome I or Chromosome II, as described previously (Petrova et al., 2013, DOI: 10.1128/MCB.01400-12). These strains also expressed a nuclear TetR-tdTomato fusion for live-cell fluorescence imaging. Figure 6—figure supplement 2 summarizes chromosome mis-segregation at Chromosomes I and II and Figure 6 describes chromosome mis-segregation at the rDNA locus (Chromosome III). In short, we find that normally dividing cells have similarly low rates of chromosome mis-segregation (7%, 5%, and 7% of cells showed defects for chromosomes I, II, and III, respectively). However, rDNA defects were elevated relative to chromosome I and II mis-segregation in cells immediately preceding death (16%, 13%, and 40% for chromosomes I, II, and III, respectively).

Chromosome I and II mis-segregation was elevated 2.3 and 2.6-fold in dying cells, whereas rDNA instability was elevated 5.7-fold. Based on these observations we conclude that rDNA instability is elevated above general chromosome mis-segregation defects. A discussion of these results is included in the last paragraph of the subsection “Ribosomal DNA (rDNA) instability contributes to sudden cell death”.

*Sir2-dependent rDNA array stabilization is considered one of the key mediators of RLS regulation in S. cerevisiae. It would be interesting to test whether Sir2 overexpression or rapamycin prevents the rDNA dynamics defects in replicatively "aging" S. pombe cells. In other words, is the rDNA contributing to the aging-independent RLS extension by Sir2 overexpression or rapamycin?*

We tackled this question by determining how rapamycin and Sir2p overexpression modulate the longevity on the *rqh1Δ* strain. The results are summarized in Figure 6—figure supplement 3 and Supplementary file 2. Rapamycin showed a mild RLS extension, whereas the *rqh1Δ sir2OE* strain was extremely short lived. These data suggest that the elevated rDNA (and likely chromosome mis-segregation) defects cannot be rescued by Sir2p and are only partially rescued by rapamycin. These findings are discussed in the last paragraph of the subsection “Ribosomal DNA (rDNA) instability contributes to sudden cell death”.

*Reviewer #2: […] The conclusions of this study are congruent with an earlier study from the lab of I. Tolic-Norrelykke (Coelho, Current Biology 2013), which also concluded that S. pombe does not age. The present work follows individual cells over a much longer period of divisions, whereas the Coelho study analysed microcolonies with up to eight cell divisions. Nonetheless, the conclusions are basically very similar.*

*This study extends the observations of Coelho by analysis of Sir2 overexpression and rapamycin, two interventions that were previously known to extend lifespan.*

*From all I can tell, the data collection and analysis is sound. The novelty lies in the extent of data collection and the statistics, but the findings support earlier conclusions. Hence, it will be a judgment call as to whether this constitutes enough novelty for publication in eLife.*

Prior to this manuscript, at least four studies have come to conflicting conclusions regarding aging in *S. pombe* (e.g., Barker and Walmsley, Yeast, 1999; Minois et al., Biogerontology, 2006; Erjavec et al., 2008, Coelho et al., Current Biology, 2013). Three manuscripts invoked aging and only one (Coelho et al) came to the opposite conclusion. These conflicting studies likely stem from the technical difficulties of monitoring individual cells that divide by binary fission.

Coelho *et al.* drew their conclusions from the analysis of six to eight cell divisions in a micro- colony. However, analyzing just eight cell divisions (~24 hrs in the lab) means that the vast majority of cells are still living. This is insufficient to draw broad conclusions about the replicative lifespan of any organism that can divide 70+ times before death (over a week of continuous division). Without a better understanding of the full replicative lifespan of *S. pombe*, which turns out to be ~40 generations long, it is also difficult to put the cell deaths observed during 8 generations in its proper context. In contrast, our study observed the full replicative lifespan of hundreds of individual cells from multiple biological replicates and a number of different genotypes. This rich dataset allowed us to develop statistical tests for aging and to fit the lifespan data to a model that separating aging dependent and aging independent factors. This significantly expands the datasets presented by Coelho et al. and definitively settles the long- standing controversy of whether fission yeast ages.

This manuscript also presents *the first high-throughput microfluidic device* for replicative lifespan studies in rod-shaped eukaryotic cells. The reviewers raise the concern that seemingly similar devices have been reported previously (see rebuttal below). However, this platform is far beyond those earlier devices, none of which could be used for replicative lifespan studies.

Briefly:

1) multFYLM is the first device that allows multiplexed observation of individual cells from six distinct strains. This allows direct comparison across genotypes in the same experiment and opens future avenues for phenotypic and fluorescent screening for aging-associated phenotypes (e.g., the gene deletion collection).

2) We can monitor thousands of *individual cells and their siblings* over their entire lifespans, frequently spanning a week of continuous observation. This is unprecedented stability for such a microfluidic platform.

3) We developed an automated and high-content image processing pipeline that facilitates rapid analysis of the massive imaging data generated using this platform. This pipeline is broadly applicable to other high-throughput microscopy studies and is freely available via the lab GitHub page (https://github.com/finkelsteinlab).

4) Finally, the device is constructed using scalable and widely available soft lithography protocols. This will allow easy adoption in many other laboratories and will facilitate scale-up to even larger integrated devices for monitoring aging.

“This study extends the observations of Coelho by analysis of Sir2 overexpression and rapamycin, two interventions that were previously known to extend lifespan.”

These interventions have only been shown to extend lifespan in model organisms that also age (e.g., budding yeast, worms, flies). Prior to our study, it was unclear whether overexpression of Sir2p or addition of rapamycin can extend lifespan without fundamentally affecting aging. Our study is the first to suggest that lifespan can be increased without any aging phenotypes, a fact that has implications for the proposed mechanism of lifespan extension by these interventions. We have included this point in the Discussion of the revised manuscript.

*Reviewer #3:*

*The authors used a microfluidic device to trap individual fission yeast cells and monitored their growth and death patterns. The main conclusion is that fission yeast does not age, in terms of growth and death rates, with respect to replicative age. While the manuscript presents interesting, solid results, the microfluidic device ("FYLM") was published by the same group previously (Spivey et al., 2014) and the conclusion that S. pombe does not age was also published by another group (Coelho et al., 2013). The major contribution of the present work is the confirmation of the latter by extending the replicative age in the experiments by 5 to 10 fold. These recent advancements are reminiscent of the development in bacterial senescence several years ago (Wang et al., 2010 and references therein).*

Please see our response to referee #2 regarding the novelty of our method and findings.

*Comments:*

*1) In Abstract, "Nearly all RLS studies have used budding yeast,.…" -> "Nearly all RLS studies of single-cell eukaryotes have used budding yeast,.…"*

*2) Also in Abstract, "Here, we describe a multiplexed fission yeast lifespan micro-dissector (FYLM); a microfluidic platform for performing high-throughput and automated single-cell micro- dissection." This sentence reads as if this is the first time the authors are reporting their microfluidic device, but it was already published in Spivey et al., 2014.*

The Abstract has been revised to:

“However, little is known about aging and longevity in symmetrically dividing eukaryotic cells because most prior studies have used budding yeast for RLS studies. Here, we describe a multiplexed fission yeast lifespan micro-dissector (multFYLM) and an associated image processing pipeline for performing high-throughput and automated single-cell micro-dissection.”

These changes highlight the novelty presented in this manuscript. Also, see our response to reviewer #2 regarding the overall novelty of this finding.

*3) The puns "FYLM", "multiFYLM", "FYLM Critic" are cute, but rather disingenuous. The Figure 1 in the manuscript and the figure in the Abstract in Spivey et al., 2014 remarkably resembles Figure 1 in Wang et al., 2010 for bacteria. Why invent a new name when the basic design of the device is essentially identical to what has been already published and widely adopted?*

Our method is different from the device published by Wang et al. (2010, DOI: DOI:10.1016/j.cub.2010.04.045) in both function and scale. First, we used standard microlithography, a technique available to nearly all biophysical and single-cell biology labs. Due to the relatively small size of *E. coli* cells, Wang et al. used an unconventional and highly specialized fabrication strategy. A second major difference is that our device has continuous media flow through the cell capture chambers (Wang et al. did not). This ensures that each cell is maintained in an environment that provides nutrients and prevents any accumulation of waste products. This was necessary because of the significantly longer timescales of our experiments and the robust growth of bacteria relative to yeast in spent media. Third, our devices can multiplex up to six distinct genotypes into a single experiment and monitor multi-color fluorescent signals throughput the replicative lifespan. Fourth, we provide both an image processing toolkit and a statistical framework for analyzing the datasets obtained via our device. Although we drew inspiration from the cartoon illustration in Wang et al., both the images and the actual platform are significantly different and highlight the novelty of our approach.

The reviewer shows concern about our nomenclature (multiplexed fission yeast lifespan microdissector; multFYLM) and suggests keeping the naming convention established in Wang et al.(“mother machine”). This terminology is unfortunately ill-fitting in regards to *S. pombe*.

Here, motherhood, (and thus the “mother machine”) is a misleading descriptor, given that *S. pombe* divides symmetrically by binary fission to produce nearly-indistinguishable cells. Moreover, the name “mother machine” pre-supposes that a mother-daughter relationship must already exist in the data, potentially biasing data analysis and interpretation. Thus, we chose to call the two new cells “siblings” and avoid mention of any mother-daughter relationship in either the device name or the discussion of our work.

*4) Figure 2. I suggest to plot from 0.1 to 1.0 for y-axis. Since the main result in this figure is the exponential decay of 'fraction surviving', the current range (0.01 to 1.0) puts unnecessary importance on what is less important.*

Figure 2 has been updated with a plot from 0.1 to 1 for the y-axis.

*5) Figure 2 and in the main text. I suggest the authors call "the hazard function" a more easily understandable and straightforward "death rate".*

“Hazard function” and “hazard rate” are related but distinct terms. The hazard function defines the functional form of the hazard rate, which boils down to two adjustable parameters in the Gompertz aging model. We agree with the reviewer that “death rate” and “hazard rate” are used interchangeably in the literature. To maximize clarity, we use “hazard rate” throughout the manuscript, but state that “hazard rate” and “death rate” are equivalent terms in the first paragraph of the subsection “The fission yeast replicative lifespan is not affected by aging”.

*6) Figure 2—figure supplement 2; Figure 5—figure supplement 2. The exponential decay and constant death rates are not as clear here. For 2B, the data is presented in a misleading manner. The range of y axis should be from 0 to 0.07. The red fit lines should not mask the data in both Figure 2 and Figure 2.*

The thickness of the red line in Figure 2 has been reduced to show a clearer view of the raw data. For maximum clarity, all RLS data relating to Figure 2, Figure 5, and Figure 6 is also included in Supplementary file 2. Supplementary file 2 including the number of cells, mean RLS (with 95% C.I.), exponential decay parameters and quality of the fits.

The range of the y-axis in Figure 2 was selected to demonstrate the drastic differences between the *S. cerevisiae* and *S. pombe* hazard functions. Truncating the y-axis to 0.07 would essentially make it impossible to see how *S. cerevisiae* ages. It would also potentially cause readers to over-interpret the noise in the *S. pombe* hazard function. However, we have matched the scaling of the y-axis on Figure 5—figure supplement 2 and Figure 6—figure supplement 3 to compare the hazard rates of wild type and *rqh1∆* strains. These parameters are also summarized in Supplementary file 2.